# Effects of Virtual Reality Interventions for Needle-Related Procedures in Patients with Cancer: A Systematic Review and Meta-Analysis

**DOI:** 10.3390/cancers17121954

**Published:** 2025-06-12

**Authors:** Jie Dong, Wenru Wang, Kennis Yu Jie Khoo, Yingchun Zeng

**Affiliations:** Alice Lee Centre for Nursing Studies, Yong Loo Lin School of Medicine, National University of Singapore, Singapore 117599, Singapore; e0556145@u.nus.edu (J.D.); e0775823@u.nus.edu (K.Y.J.K.); chloezengyc@hotmail.co.uk (Y.Z.)

**Keywords:** virtual reality, needle-related procedures, patients with cancer, psychological symptoms, systematic review and meta-analysis

## Abstract

Patients with cancer often face pain, anxiety, fear, and distress during needle-related procedures, which can impact their well-being. This study reviewed 14 clinical trials with over 1000 participants to assess whether virtual reality can ease these symptoms. Findings showed that virtual reality was effective in reducing pain, anxiety, depression, fear, and respiratory rate in both adults and children. However, it did not significantly affect heart rate. By offering calming and engaging experiences, virtual reality helped distract patients and reduce their discomfort. These results suggest that virtual reality is a promising non-drug option to improve patient comfort during cancer care. Further research is needed to refine and expand its use across diverse settings and populations.

## 1. Introduction

Cancer is one of the leading causes of global mortality, with its prevalence expected to rise significantly in the coming decades, reaching an estimated 28.4 million cases by 2040 [1]. As survival rates improve, patients with cancer often face diminished quality of life (QOL) due to the psychological and physiological impacts of the disease and its treatments. Common issues such as anxiety, depression, pain, and fear are prevalent, with many patients also experiencing significant distress during needle-related procedures (NRPs), which are routine in cancer care [2]. These procedures, including bone marrow biopsies, venipuncture, and chemotherapy infusions, can induce both physical pain and psychological distress, exacerbating anxiety and fear and further diminishing patients’ QOL [3].

Although traditional pharmacological treatments, such as opioids and benzodiazepines, are commonly used to manage pain and anxiety, they often have significant side effects, limiting their effectiveness and applicability in addressing the broader emotional and psychological dimensions of these symptoms [4]. Furthermore, standard approaches to pain management do not always provide sufficient relief for the distress caused by NRPs, with studies showing that nearly 50% of patients with cancer experience inadequate pain management [5]. Consequently, there is a growing need for alternative, non-pharmacological interventions that can address the multidimensional nature of pain and psychological distress in patients with cancer undergoing NRPs.

Virtual reality (VR) has emerged as a promising solution for managing both the physical and psychological symptoms associated with cancer treatments. By immersing patients in interactive, 3D virtual environments, VR serves as a powerful distraction tool that can reduce the cognitive resources available for pain processing [6]. VR has been shown to alleviate pain [6,7], anxiety [2], depression [7], and fear [8], providing both psychological and physiological benefits. Moreover, VR-based interventions can be tailored to individual needs, offering different levels of immersion—ranging from full-immersive systems with head-mounted display (HMD) to non-immersive systems that utilize smartphones or computers—depending on the therapeutic context [9]. Non-immersive VR uses a desktop screen with keyboard mouse to navigate 3D environments. Semi-immersive VR projects the environment onto a large screen and may include gesture recognition. Fully immersive VR uses an HMD to block visual and auditory distractions, creating a highly immersive experience [9]. Non-immersive VR has shown benefits in adult rehabilitation by supporting physical fitness and balance [10], while immersive VR is more commonly used in pediatric settings, where children’s sensitivity to medical cues makes them more receptive to immersive distraction [11].

Despite the growing body of evidence on the efficacy of VR in managing cancer-related symptoms, there remain significant gaps in research focusing specifically on patients with cancer undergoing NRPs. While previous systematic reviews (SR) have examined VR’s effectiveness in either patients with cancer or patients undergoing NRPs [11,12,13,14], they present key limitations, highlighting the need for the present SR to address these gaps.

Three recent SRs [11,12,13] focused exclusively on pediatric and adolescent populations, neglecting adults despite their increasing cancer burden, as older adults (>65 years) will account for 60% of newly diagnosed cancer cases by 2035 [15]. Additionally, young adult patients with cancer (<39 years) experience worse quality of life and greater psychosocial burdens compared to pediatric patients with cancer [16], underlining the need for broader population coverage. Moreover, the generalizability of findings from these three SRs’ is limited by VR interventions designed for children, such as animal-themed environments [17], ocean rift simulations [17,18], and exergames [19]. These child-oriented VR applications may not be suitable for adults, who may benefit more from nature-based VR interventions shown to improve mood and reduce pain [20,21]. Thus, this SR will include both adult and pediatric oncology populations, with a subgroup analysis to evaluate VR’s differential impacts between these groups.

Meanwhile, Gautama et al. [14] included adult patients with cancer in their review, but they focused primarily on chemotherapy-related symptoms, overlooking the broader spectrum of routine NRPs in oncology, including venipuncture, bone marrow biopsies, and lumbar puncture. Furthermore, while the review showcased VR’s ability to reduce anxiety and pain, it did not differentiate between chemotherapy-induced and procedural-related discomforts, which have distinct underlying mechanisms. Chemotherapy-induced pain and anxiety result primarily from neurotoxic effects of chemotherapeutic agents [22,23], whereas procedural pain arises from curative surgery or short-lived NRPs [24,25]. By expanding beyond chemotherapy, this SR evaluates VR’s effectiveness across various NRPs, addressing an important yet understudied aspect of cancer symptom management.

Previous SRs also had methodological limitations. Three of the aforementioned SRs included fewer than ten studies with small sample sizes [11,12,13], limiting their findings’ generalizability and external validity. In contrast, Gautama et al. [14] included twelve studies, but one-third were published eighteen years ago, making their findings less applicable given the rapid VR advancements and the availability of more up-to-date randomized controlled trials (RCTs) [26]. Additionally, Tsitsi et al. [11] did not perform a meta-analysis, instead emphasizing VR’s usability over its effectiveness. Their narrative synthesis limits the translation of findings into clinical practice.

Therefore, this SR and meta-analysis aims to evaluate the effectiveness of VR-based interventions in reducing pain, anxiety, depression, fear, pulse rate, and respiratory rate in patients with cancer receiving NRPs. By collating and analyzing the latest RCTs, this review seeks to address the under-explored area of VR interventions in cancer care and provide evidence for integrating VR as a complementary tool in managing the complex symptoms associated with NRPs in oncology patients.

## 2. Methods

This review was conducted in accordance with the Preferred Reporting Items for Systematic Reviews and Meta-Analyses (PRISMA) guidelines [27]. A formal protocol was registered prospectively in PROSPERO under registration ID CRD42025615497. The PRISMA 2020 checklist was presented in Appendix A. 

### 2.1. Search Strategy

Two independent reviewers employed a three-step search strategy across eleven databases: CINAHL, Cochrane, Embase, IEEE Xplore, Medline, ProQuest, PsycINFO, and PubMed. Additionally, three grey literature sources—GreySource, OpenGrey, and Google Scholar—were consulted. The search period spanned from August 2024 to May 2025. An experienced school librarian validated the search strategy. As all reviewers were proficient in both English and Chinese, studies in both languages were included. Key search terms included ‘Virtual reality,’ ‘Head-mounted display,’ ‘needle,’ ‘venipuncture,’ ‘cancer,’ and ‘oncology.’ Detailed search strategies for each database are provided in Appendix A.

### 2.2. Eligibility Criteria

The inclusion criteria were structured using the Population, Intervention, Comparison, and Outcome (PICO) framework:
Population: Patients with cancer receiving NRPs, regardless of age or cancer type, including both active treatment and survivor groups.Intervention: Any type of VR-based intervention (immersive, semi-immersive, or non-immersive), irrespective of setting, duration, session frequency, or application methods.Comparison: Usual care, which may include standard medical treatment, psychosocial care, or health education.Outcomes: Anxiety, pain, depression, fear, pulse rate, and respiratory rate.

Only RCTs were included. Non-experimental, quasi-experimental studies, reviews, conference papers, and other non-RCT designs were excluded. Moreover, languages other than English and Chinese were excluded from the study. A detailed summary of the eligibility criteria is provided in Appendix A.

### 2.3. Study Selection

Search results were organized and imported into EndNote X9.3.3 software. After duplicates were removed, two independent reviewers (DJ and KYJ) cross-screened the titles and abstracts of the remaining studies. Discrepancies were resolved by consulting with two additional reviewers, WWR and ZYC. Full texts of eligible studies were retrieved for further evaluation, with reasons for exclusion recorded in the PRISMA flow chart.

### 2.4. Data Extraction

A standardized data extraction form was adapted based on the Cochrane Handbook for Systematic Reviews of Interventions [28]. A pilot data extraction was conducted on five studies by two reviewers (DJ and KYJ) to ensure consistency and relevance. Extracted data included authorship, publication year, country, sample size, mean age, cancer type, NRP details, VR methods, VR duration, outcomes, measurement tools, and main results. Any disagreements were resolved with input from WWR and ZYC.

### 2.5. Risk of Bias and Quality Appraisal

The quality of included studies was assessed independently by two reviewers using the Cochrane Risk-of-Bias (ROB) Tool version 1 [29]. The assessment evaluated five types of biases: selection bias, performance bias, detection bias, attrition bias, and reporting bias, across seven domains: random sequence generation, allocation concealment, blinding of participants and personnel, blinding of outcome assessment, incomplete outcome data, selective reporting, and other biases. Each domain was rated as ‘low risk,’ ‘high risk,’ or ‘unclear.’ Discrepancies were resolved through discussions with both reviewers, WWR and ZYC. The overall quality of evidence was evaluated using the GRADE framework [28], with ratings ranging from high to very low based on five domains: risk of bias, inconsistency, indirectness, imprecision, and publication bias. Each outcome was rated separately using GRADEpro software [30], with inter-rater agreement calculated using Cohen’s Kappa (κ).

### 2.6. Data Synthesis and Analysis

Meta-analysis was conducted using the online Cochrane Review Manager (RevMan) [31]. All outcomes were treated as continuous data, with the standardized mean difference (SMD) and mean difference (MD) reported with 95% confidence intervals. SMD was used for outcomes measured with different tools, while MD was applied for consistent measurement tools [32]. Effect sizes were categorized as small (<0.5), moderate (0.5–0.8), or large (>0.8) using Cohen’s definitions [28]. Heterogeneity among studies was assessed using I^2^ statistics and Cochran’s Q chi-square test. I^2^ values of 0–40%, 30–60%, 50–90%, and 75–100% corresponded to unimportant, moderate, substantial, and considerable heterogeneity, respectively. For low heterogeneity (*p* > 0.1 and I^2^ < 50%), a fixed-effect model was applied [33]. In cases of significant heterogeneity (*p* ≤ 0.1 and I^2^ ≥ 50%), a random-effects model was used. Subgroup analyses were performed to explore sources of high heterogeneity.

## 3. Results

A total of 8027 references were retrieved from the 11 databases, with an additional 3 records identified through hand-searching Google Scholar. Figure 1 illustrates the study section process.

### 3.1. Characteristics of Included Studies

A total of 14 included studies were conducted in the following countries: China (*n* = 4) [34,35,36,37], Germany (*n* = 1) [38], Italy (*n* = 1) [39], Spain (*n* = 1) [40], and Turkey (*n* = 7) [18,41,42,43,44,45,46]. Sample sizes ranged from 19 to 139 participants [35,43]. In these studies, seven studies involved children [18,35,36,38,41,44,45], while another seven involved adults [34,37,39,40,42,43,46].

Regarding types of NRPs that participants receiving, five studies focused on chemotherapy infusion [35,37,39,40,46], three studies on venipuncture and venous blood drawing [36,41,45], four studies on port needle insertion [18,38,43,44], one study on bone marrow aspiration [42], and one study on radical mastectomy [34]. The VR interventions varied in equipment types (headset or smart glasses), duration (8 min to 45 min or procedure-long), frequency (single or multiple sessions), and content (natural scenes or psychoeducational modules). Detailed characteristics of each study are shown below (Table 1).

### 3.2. Quality and Risk of Bias Assessment

All studies sufficiently described random sequence generation, ten studies mentioned allocation concealment [35,36,37,39,40,42,43,44,45,46], and six studies adopted blinded outcome assessments [35,37,39,40,42,43]. Due to the nature of VR-based interventions, participant and personnel blinding were not feasible. Additionally, thirteen studies exhibited low risk of attrition bias, with one study having a high risk due to 11% dropout rate in the VR intervention group [41]. All fourteen studies had a low risk of reporting bias. The risk of bias graph and summary are shown in Figure 2. The two independent reviewers had a substantial inter-rater agreement with Cohen’s kappa value 0.74 [47].

According to the GRADE approach, the quality of evidence for all outcomes was graded very low due to methodological limitations, significant heterogeneity, and small sample sizes. Further details are presented in Appendix A.

### 3.3. Effectiveness of VR Interventions on Primary and Secondary Outcomes

The effects of VR-based interventions on anxiety were measured in 12 studies involving 949 participants [18,34,35,36,37,38,39,40,42,43,44,46]. Results demonstrated that VR-based interventions significantly improved anxiety symptoms in patients with cancer undergoing NRPs compared to control groups (SMD = −1.74, 95% CI −2.47 to −1.01, *p* < 0.001) (Figure 3a). In the subgroup analysis by age (adult and children), the results showed that VR-based interventions significantly improve anxiety symptoms in adult or children patients with cancer receiving NRPs compared to control groups (SMD = −2.12, 95% CI −3.27 to −0.97, *p* = 0.0003; SMD = −1.18, 95% CI −1.87 to −0.48, *p* = 0.0009, respectively) (Figure 3b).

The effectsf VR-sed interventions on pain were measured in eight studies involving 638 patients [18,36,38,41,42,43,44,45]. The results indicated that VR-based interventions significantly reduced pain compared to control groups (SMD = −1.30, 95% CI −1.93 to −0.67, *p* < 0.001) (Figure 4a).

In the subgroup analysi [18,36,38,41,42,43,44,45]. by age (adult and children), the results revealed that VR-based interventions significantly improved pain symptoms in both adults (SMD = −1.57, 95% CI −3.00 to −0.13, *p* = 0.03) and children (SMD = −1.12, 95% CI −1.17 to −0.53, *p* = 0.0002) (Figure 4b).

The effects of VR-based interventions on depression in patients with cancer were measured in three studies including 313 patients [34,37,40]. VR-based interventions significantly reduced depressive symptoms in patients with cancer receiving NRPs (SMD = −0.73, 95% CI −0.96 to −0.50, *p* < 0.001) (Figure 5). Fear-related outcomes were measured in three studies with 173 patients [18,41,44]. The results showed that VR-based interventions effectively improved fear in patients with cancer compared to control groups (MD = −1.31, 95% CI −1.56 to −1.06, *p* < 0.001) (Figure 6).

Three trials reported the results of VR-based interventions on pulse rate [35,36,43]. Results showed no statistically significant effect between VR and control groups (MD = 0.25, 95% CI −14.32 to 14.81, *p* = 0.97) (Figure 7a). Conversely, two trials involving 201 patients reported that VR was effective in reducing respiratory rate compared to control groups (MD = −3.85, 95% CI −6.18 to −1.52, *p* = 0.001) [43,44] (Figure 7b).

## 4. Discussion

This [35,36,43] review examined the efficacy of VR-based interventions in patients with cancer receiving non-pharmacological treatments (NRPs). The results demonstrate that VR-based interventions significantly reduced anxiety, pain, depression, fear, and respiratory rate compared to standard care. These findings align with previous reviews emphasizing VR’s effectiveness in alleviating anxiety, pain, depression, and fear [2,7,8]. However, these results differ from another review [48], where no significant effects were found. This discrepancy may be attributed to differences in sample sizes, VR intervention types, session durations, and population characteristics.

The effect of VR on pulse rate could not be conclusively determined due to statistical insignificance, which aligns with a recent systematic review indicating no impact of VR-based interventions on pulse rate in cancer symptom management [14]. Therefore, more robust RCTs are needed to explore the impact of VR on pulse rate among patients with cancer undergoing NRPs. Overall, these findings highlight the psychological and physiological benefits of VR, offering valuable insights into its broad potential.

### 4.1. Effects of VR on Psychological Outcomes of Patients with Cancer

Anxiety, depression, and fear are common psychological symptoms among patients with cancer, particularly in children and adolescents [49]. These symptoms often stem from pain associated with NRPs and the inherent uncertainties of the disease [50,51]. Anxiety is particularly prevalent in patients undergoing chemotherapy [52]. Of the 14 trials included in this review, five focused on chemotherapy patients, with treatment durations ranging from single 20–40 min [35,39] to multiple daily sessions [37,40,46]. Chemotherapy-induced anxiety often arises from uncertainties regarding treatment outcomes, potential pain, side effects, and efficacy [40]. The efficacy of VR in alleviating chemotherapy-related anxiety likely lies in its ability to alter a patient’s perception of time, making the experience seem shorter and more tolerable [40]. Additionally, two studies incorporated specialized psychoeducation modules into VR interventions [34,40], and one study integrated mindfulness meditation [37], both of which are proven to effectively reduce anxiety [53,54].

For shorter-duration NRPs, such as blood drawing, venipuncture, lumbar puncture, and port insertion, VR served as an effective distraction. By diverting the patient’s attention and immersing them in calming, engaging virtual environments, VR alleviated anxiety and provided a temporary escape from the clinical setting [21]. Moreover, subgroup analysis revealed that VR’s effectiveness in reducing anxiety was consistent across both children and adults, suggesting that VR can be effectively used across age groups.

Regarding VR’s ability to reduce fear, all three studies involving children demonstrated its efficacy in alleviating intense fear and distress [55]. Fear in the context of cancer arises from uncertainty about the disease, loss of control over its management, and fear of mortality [56]. In children, this fear is often exacerbated by the discomfort associated with needles and pain [57]. Without appropriate interventions, such fear can lead to negative emotional trauma and avoidance of future treatments [58,59]. VR’s ability to engage children with fascinating cartoons and interactive VR tools effectively distracted them from needles, reducing their fears. This underscores VR’s potential as a powerful non-pharmacological intervention for pediatric patients’ psychological needs during cancer treatment.

Although only three trials showed that VR-based interventions reduced depression, several factors likely contributed to this effect. Two studies incorporated natural themes and auditory elements into the VR experiences [37,40], with virtual immersion in natural environments—such as forests and flowers—paired with calming auditory cues like flowing water and birdsong, shown to improve mood and reduce blood pressure [60]. Interaction with nature through VR has been linked to increased psychological resilience and reduced depression [20,61,62]. Additionally, all three studies incorporated meditation or psychoeducation programs, providing participants with tools to achieve calmness, further contributing to the reduction in depressive symptoms [63,64,65].

### 4.2. Effects of VR on Physiological Outcomes of Patients with Cancer Aspects

The review found that VR-based interventions effectively reduced pain and respiratory rate, consistent with multiple reviews [6,7,66]. Cancer pain arises from tumor involvement, anticancer treatment, and painful NRPs such as venipuncture and lumbar puncture [67]. These painful experiences are often intensified by fear and anxiety, exacerbating physiological responses like increased respiratory rate and pain perception [6]. Physiologically, VR reduces adrenergic activity by eliciting an anxiolytic effect and suppressing pain signals [42]. The gate control theory [68] further explains VR’s efficacy, as immersive content captures attention, reducing the capacity for pain processing. In our analysis, VR interventions consistently diverted attention from painful stimuli, leading to significant pain relief. Subgroup analysis revealed no significant differences in VR effectiveness between children and adults, suggesting that VR’s mechanism of action operates uniformly across age groups. This echoes the findings of Niaz et al. [69], who reported VR’s effectiveness in reducing procedural pain in children, possibly due to their vivid imagination and greater pain sensitivity, though its impact on adults was not addressed. Moreover, pain reduction also led to a corresponding decrease in respiratory rate, reflecting their closely linked bidirectional relationship [70].

Surprisingly, VR had no effect on pulse rate, which contrasts with findings from previous reviews [6,71]. This discrepancy may be due to differences in the types of NRPs studied, such as port catheter implantation [43], intravenous cannulation [36], and chemotherapy [35], each of which triggers varying pain responses and autonomic reactions [72,73]. The timing and duration of pulse rate measurements during these procedures could also have influenced the outcomes. Future studies should address these limitations by increasing the frequency of pulse rate measurements, using manual methods for accuracy, and recording pulse rate over a full minute.

### 4.3. Study Strengths and Limitations

This is the first meta-analysis to examine the effectiveness of VR-based interventions on patients with cancer receiving NRPs. With NRPs being a common occurrence in cancer care, this study focuses on a specific and underserved subset of the cancer population. The inclusion of only RCTs ensured robust causal–effect relationships, and randomization minimized potential biases [74]. The review included studies in both English and Chinese, offering a more comprehensive perspective. The inclusion of studies from 2021 to 2024 ensured the evidence is relevant to current healthcare practices. Furthermore, the study demonstrated the broad applicability of VR interventions, with diverse scenarios—from immersive natural environments to psychoeducational modules—tailored to meet various patient needs.

However, controlling heterogeneity was challenging due to variations in VR types, themes, durations, and session frequencies, as well as the mixed adult and pediatric populations. Additionally, the nature of VR-based interventions made it difficult to blind participants, potentially influencing subjective outcomes like pain and anxiety. Some studies also included psychological interventions in the control groups, which could have confounded results. Furthermore, only one Chinese article was included, possibly due to limitations in the authors’ search strategies. Previous reviews have highlighted potential downsides of VR interventions, such as discomforts related to immersive VR using head-mounted displays, including cybersickness and digital eye strain [7,75]. Given that patients with cancer often experience additional discomforts, further research is needed to explore these side effects and their implications. Additionally, the limited number of studies on depression, fear, and physiological parameters may affect the reliability and precision of the findings.

### 4.4. Implications and Future Research

These findings highlight that VR-based interventions could be a valuable adjunct or alternative for alleviating symptoms in patients with cancer undergoing NRPs, with potential for integration into healthcare services as a non-pharmacological intervention. It is recommended that VR training be incorporated into nursing curricula and hospital training programs, with multi-disciplinary collaboration involving psychologists, health informatics teams, rehabilitation specialists, and oncology providers to optimize its implementation. Future research should focus on determining the optimal duration, frequency, and content of VR interventions, while also exploring long-term benefits, personalized content, and integration with other therapeutic treatments like cognitive–behavioral therapy or mindfulness. Additionally, addressing accessibility, cost, and scalability will be crucial for ensuring equitable access to VR interventions across diverse healthcare settings.

## 5. Conclusions

This review evaluated the effectiveness of VR-based interventions in alleviating psychological and physiological symptoms in patients with cancer undergoing needle-related procedures (NRPs). VR significantly reduced anxiety, pain, depression, fear, and respiratory rate, demonstrating its potential as a valuable adjunct to traditional cancer care. The use of immersive scenes, cartoons, psychoeducation, and mindfulness programs addresses both mental and physical distress, supporting VR’s integration into clinical practice. Yet, limitations such as high heterogeneity of interventions, limited studies on depression and physiological parameters, and variability in NRP types need to be addressed. Future research should focus on personalized VR content, assess potential side effects like cybersickness, and tailor interventions to different NRPs to optimize patient outcomes.

## Figures and Tables

**Figure 1 cancers-17-01954-f001:**
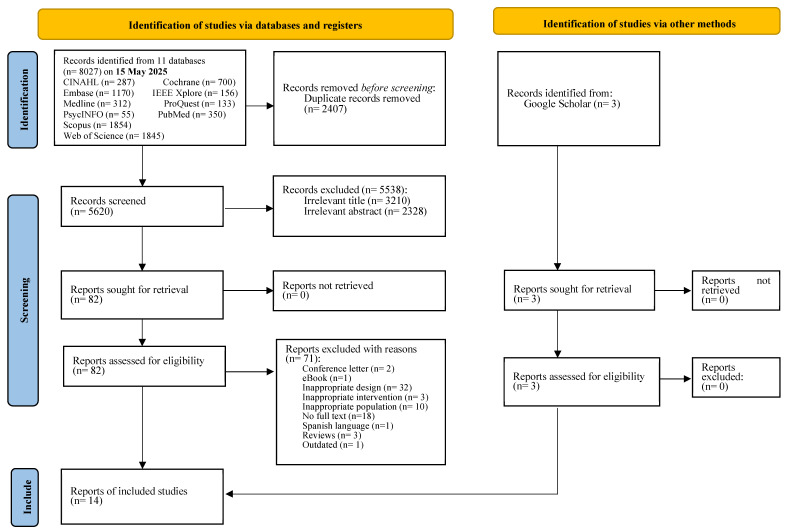
PRISMA flow chart.

**Figure 2 cancers-17-01954-f002:**
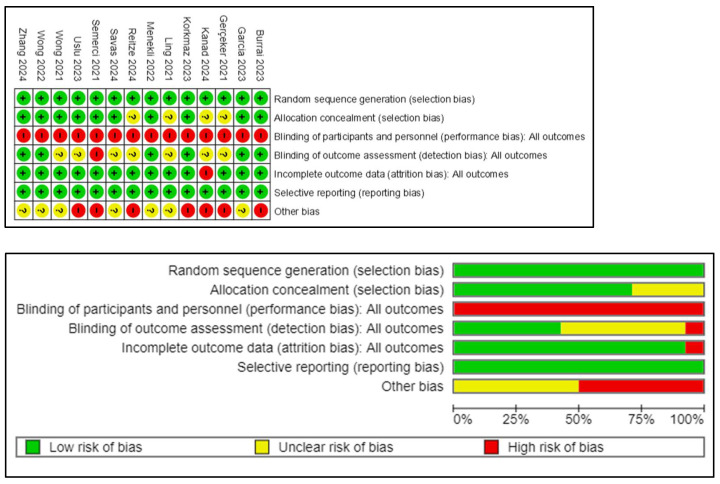
Risk of bias analysis of the included studies [18,34,35,36,37,38,39,40,41,42,43,44,45,46].

**Figure 3 cancers-17-01954-f003:**
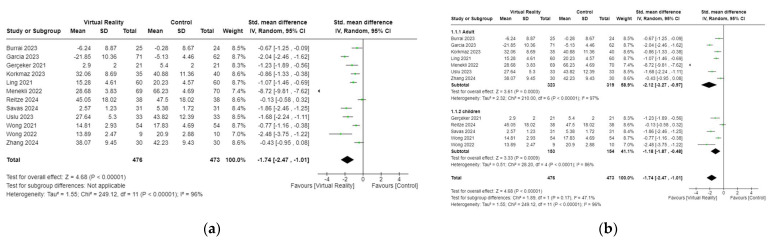
(**a**) Forest plot of VR on anxiety. (**b**) Subgroup analyses of VR on anxiety for patients of different age among patients with cancer [18,34,35,36,37,38,39,40,42,43,44,46].

**Figure 4 cancers-17-01954-f004:**
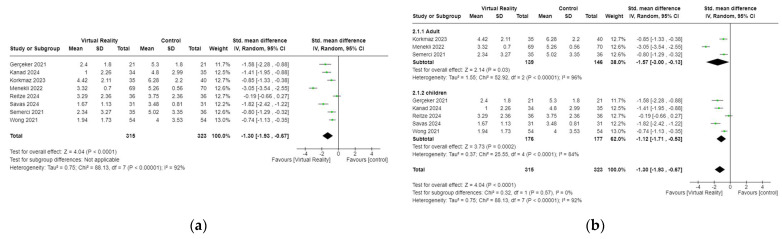
(**a**) Forest plot of VR on pain. (**b**) Subgroup analyses of VR on pain for patients of different age among patients with cancer [18,36,38,41,42,43,44,45].

**Figure 5 cancers-17-01954-f005:**
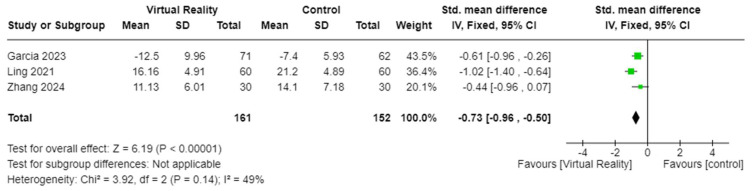
Effectiveness of VR on depression among patients with cancer [34,37,40].

**Figure 6 cancers-17-01954-f006:**
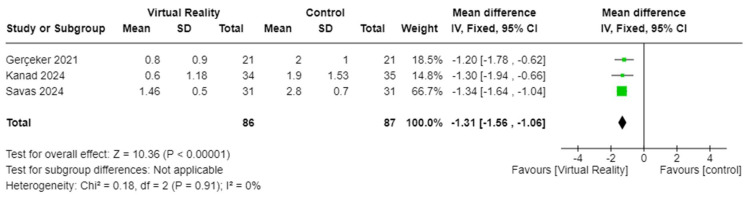
Effects of VR on fear among patients with cancer [18,41,44].

**Figure 7 cancers-17-01954-f007:**
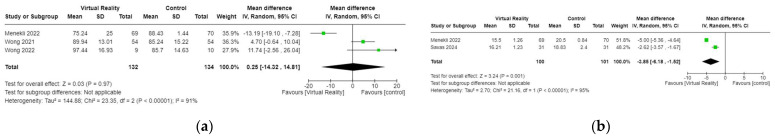
Effects of VR on physiological parameters: (**a**) Pulse rate and (**b**) Respiratory rate among patients with cancer [35,36,43,44].

**Table 1 cancers-17-01954-t001:** Characteristics of included 14 studies.

Author, Year	Country	N (E/C)	Mean Age (E/C)	Cancer Type	Needle-Related Procedure	VR Methods	VR Duration	Outcome	Outcome Instrument	Main Results
Gerçeker et al., 2021 [18]	Turkey	21/21	11.2 ± 3.1/11.7 ± 3.2	Haematology-Oncology	Port needle insertion	Standard care with wear Samsung Gear Oculus headset, to choose one of the three applications: (1) swimming with marine animals (Ocean Rift), (2) riding a rollercoaster (Rilix VR), and (3) exploring the forest through the eyes of woodland species (In the eyes of animal).	2–3 min prior and last throughout the procedure	Anxiety Fear Pain	CAM-S CFS WBS	VR is an effective distraction method in reducing port needle-related pain, fear, and anxiety in Paediatric Haematology-Oncology patients.
Ling 2021 [34]	China	60/60	53.02 ± 3.81/52.58 ± 3.78	Breast cancer	Radical mastectomy of breast cancer	Standard care with psychoeducation through VR prior and during operation	Throughout the procedure	Anxiety Depression	HAMA HAMD	VR is effective in reducing anxiety and depression during radical mastectomy of breast cancer.
Wong et al., 2022 [35]	Hong Kong	9/10	10.33 ± 1.5/9.11 ± 1.6	Cancer	First intravenous Chemotherapy	Standard care with 3 sessions of immersive VR by wearing Google cardboard goggles; 4 types of cartoon videos were provided. Three sessions: 4 h before chemotherapy administration, 5 min before and during the first chemotherapy and 5 min before and during the second chemotherapy.	41 min	Anxiety Heart rate MABP	CSAS-C Omron HBP-1100 deviceOmron HBP-1100 device	VR is effective in managing anxiety in paediatric patients receiving their first chemotherapy.
Wong et al., 2021 [36]	Hong Kong	54/54	10.2 ± 3.5/10.5 ± 3.8	Cancer	Peripheral intravenous cannulation	Standard care with wearing Google cardboard; four types of VR animated videos were allowed for participants to choose, incl. cartoons, museum, and water worlds.	5 min prior and last throughout the procedure	Anxiety Pain Pulse rate	CSAS-C VAS Pulse-Oxygen monitor	VR is effective in alleviating pain and anxiety among paediatric patients with cancer undergoing peripheral intravenous cannulation.
Zhang et al., 2024 [37]	China	30/30	33.5 ± 11.1/35.27 ± 10.6	Acute leukaemia	Induction Chemotherapy	Usual care with head-mounted display device, VR-based mindfulness meditation program. From day 1 to 14, participants experienced a total of 14 different VR 3D videos, in which they were simulated sitting in the forest or by the beach, following the meditation guide sound to meditate, listening to real, natural sounds and peaceful background music to keep the mind calm.	20 min daily for 14 days	Anxiety Depression	SAI CES-D	VR is effective in alleviating anxiety among patients during induction chemotherapy.
Reitze et al., 2024 [38]	Germany	38/38	11.5 ± 12/11.5	Cancer	Puncture procedures (port puncture and lumbar puncture)	Standard care with VR goggles to watch passive distraction videos.	Throughout the procedure	Anxiety Pain	mYPAS-SF FPS-r/NRS	VR can be used for peri-interventional reduction in pain and anxiety.
Burrai et al., 2023 [39]	Italy	25/24	62.56 ± 10.08/58.80 ± 11.54	cancer	Antineoplastic Therapy	Standard care with wear Oculus Quest 2 Head Mounted Device to watch virtual scenarios; scenarios include 9 categories: (1) Africa, (2) hills, (3) rivers, lakes, and waterfall, (4) islands, (5) deserts, (6) beaches, (7) mountains, (8) sea, (9) submarines. The scenarios have a background of nature sounds and soothing music.	30 min	Anxiety Fatigue	STAI PFS	VR is effective to reduce anxiety and fatigue in people with cancer undergoing antineoplastic therapy.
García et al., 2023 [40]	Spain	71/62	49 ± 11.7/50 ± 11.6	Breast cancer	Chemotherapy	Standard care and four sessions of VR in which a Samsung Gear device was used with a computer program developed by Psious, three psychoeducational modules (module 1: seabed virtual environment to allow relaxation, module 2: seabed with different marine species to create a mindfulness environment, module 3: spring and summer scenes); the modules were repeated for each session	4 sessions, each session 30–45 min	Anxiety Depression Emotional distress	HAD HAD DME	VR can improve anticipatory anxiety, depression, and emotional distress in breast patients with cancer who must start chemotherapy.
Kanad et al., 2024 [41]	Turkey	34/35	8.2 ± 2.72/8 ± 2.73	Haematology-Oncology	Blood draw	Standard care with Oculus Quest 2256 GB All-In-One VR goggles were used, and the Epic Roller Coasters application was selected as the VR application.	2 min prior and last throughout the procedure	Fear Pain	CFS WBS	VR is an effective approach to reducing the negative emotional appearances and for relieving pain and fear in children aged 4–12 years undergoing blood-draw procedure.
Korkmaz & Guler 2023 [42]	Turkey	35/40	49.86 ± 15.63/50.18 ± 16.26	Haematology- Oncology	Bone Marrow Aspiration	VR goggle was placed over participant’s eyes during the procedure, and one of the relaxing videos with nature sounds including underwater, museum, park, and hiking images were shown, based on participant’s preference.	15–20 min	Anxiety Pain	STAI VAS	Video streaming with VR goggle reduces pain and anxiety felt by adult patients during bone marrow aspiration procedure.
Menekli et al., 2022 [43]	Turkey	69/70	20–63	Cancer	Port catheter implantation	Turkcell T-VR glasses were used with smartphones to provide visual and auditory experience for participants undergoing port implantation procedure; varied number of parks, nature and seaside walks, submarine, museum tours with thematic music were used, each lasting approximately 3–10 min, and participants could choose the video they wanted to watch.	45 min	Anxiety Pain Heart rate Respiration	SAI VAS Con-Tec CMS5100 device Con-Tec CMS5100 device	VR is effective in reducing pain, anxiety, systolic blood pressure, diastolic blood pressure, heart rate, and respiratory rate and increase the Spo2 of the patients undergoing port catheter implantation.
Savaş et al., 2024 [44]	Turkey	31/31	9.33 ± 2.08/9.74 ± 1.76	Cancer	Port needle insertion	Standard care with VR headset and the breathing sensor applied over chest; there were two scenario-based video games that user could choose and actively interact with by breathing; scenarios were submarine and garden, and user needed to breath regularly for VR game to progress.	6 min	Anxiety Fear Pain Respiration	CASS CFS WBS ADXL354 sensor	The VR reduces procedure-related pain, anxiety and fear. The VR has a positive effect on the mean respiratory rate.
Semerci et al., 2021 [45]	Turkey	35/36	11.69 ± 36.6/11.67 ± 3.55	Cancer	Venous port insertion	The Piranham^TM^ VR headset connected to an iPhone 6 mobile phone; participant was allowed to watch and listen to a roller-coaster VR video.	8 min	Pain	WBS	VR is effective for reducing pain during venous port access
Uslu & Arslan 2023 [46]	Turkey	33/33	53.42 ± 9.25/51.42 ± 8.53	Breast cancer	Adjuvant chemotherapy	Standard care with Zore G04BS VR Shinecon glasses used with smartphones; VR content included three beach and three nature options; participant was allowed to choose one of six videos	4 sessions, each session 30 min	Anxiety Fatigue	STAI CFS	VR glasses is effective in reducing anxiety and fatigue scale scores in patients with breast cancer;

Note: C = Control Group; CAM-S = The Children’s Anxiety Meter-State; CASS = Child Anxiety Scale-State; CES-D = Centre for Epidemiological Studies Depression Scale; CSAS-C = The short form Chinese version of the State Anxiety Scale for Children; CFS = The Child Fear Scale; Cancer Fatigue Scale; DME = Emotional Discomfort Detection Scale; E = Experiment Group; FPS-r/NRS = Faces Pain Scale-Revised/Numerical Rating Scale; HAMA = Hamilton Anxiety Rating Scale; HAMD = Hamilton Depression Rating Scale; MABP = Mean Arterial Blood Pressure; mYPAS-SF = The modified Yale Preoperative Anxiety Scale-Short Form; N = Sample Size; PFS = Piper Fatigue Scale; SAI = State Anxiety Inventory; STAI = State Trait Anxiety Inventory; VAS = Visual Analogue Scale; VR = Virtual Reality; WBS = Wong-Baker FACES Pain Rating Scale.

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
