# Peer review of "Effects of Virtual Reality Interventions for Needle-Related Procedures in Patients with Cancer: A Systematic Review and Meta-Analysis"

_cancers, 2025, doi:10.3390/cancers17121954_

Round 1

Reviewer 1 Report

Comments and Suggestions for Authors

This manuscript presents a timely and important systematic review and meta-analysis evaluating the effectiveness of virtual reality (VR) interventions during needle-related procedures (NRPs) among cancer patients. Given the distressing nature of NRPs and the limitations of pharmacological management, this review fills a meaningful gap in oncology symptom management by highlighting the potential of VR as a non-pharmacologic tool.

The methodology is rigorous, the search strategy is comprehensive (including 11 databases), and the inclusion of both adult and pediatric populations adds value. The subgroup analyses by age are particularly noteworthy and should be retained.

Only a few minor issues to be improved: 
1. Consider discussing the practical implications of differential VR effectiveness between adults and children. The following recent systematic review may serve as a useful reference in this context:
https://www.sciencedirect.com/science/article/pii/S266700972300026X
2. To promote respectful and non-stigmatizing language, we recommend replacing terms such as “cancer patients” with “patients with cancer” throughout the manuscript, in alignment with best practices in person-centered healthcare communication.
3. Correct minor issues such as “8,027references” (Page 5, Line 185).
4. Consider adding a glossary or table of acronyms to support comprehension for readers unfamiliar with scales like STAI or CFS.

Overall, the paper is well-written and addresses an important clinical issue. Minor improvements in discussion clarity, consistency of presentation, and language use will enhance the manuscript’s clarity and inclusivity. Well done

Author Response

#Reviewer 1

This manuscript presents a timely and important systematic review and meta-analysis evaluating the effectiveness of virtual reality (VR) interventions during needle-related procedures (NRPs) among cancer patients. Given the distressing nature of NRPs and the limitations of pharmacological management, this review fills a meaningful gap in oncology symptom management by highlighting the potential of VR as a non-pharmacologic tool.

The methodology is rigorous, the search strategy is comprehensive (including 11 databases), and the inclusion of both adult and pediatric populations adds value. The subgroup analyses by age are particularly noteworthy and should be retained.

Only a few minor issues to be improved: 

  1. Consider discussing the practical implications of differential VR effectiveness between adults and children. The following recent systematic review may serve as a useful reference in this context:
    https://www.sciencedirect.com/science/article/pii/S266700972300026X

A1: Thank you for your suggestion. We have supplemented the discussion on the differential effectiveness of VR between children and adults, with reference you kindly provided. We believe this addition enhances the depth of our discussion. Please see the revised manuscript Line 111-113.

  1. To promote respectful and non-stigmatizing language, we recommend replacing terms such as “cancer patients” with “patients with cancer” throughout the manuscript, in alignment with best practices in person-centered healthcare communication.

A2: Thank you for your suggestion. We sincerely apologize for the insufficient consideration about person-centered healthcare communication. Therefore, we have revised all “cancer patients” with “patients with cancer” throughout the manuscript.

  1. Correct minor issues such as “8,027references” (Page 5, Line 185).

A3: Thank you for your kind reminder. We have corrected the typo by inserting a space between “8027” and “references”, please see the revised manuscript Line 246.

  1. Consider adding a glossary or table of acronyms to support comprehension for readers unfamiliar with scales like STAI or CFS.

A4: Thank you for your helpful suggestion. We acknowledge that a glossary or table of acronyms would better facilitate reader’s understanding of measurement scales in our manuscript. In addition to a table footnote we previously added (see revised manuscript page 14), we have also created a list of abbreviation, which can be found in the Appendix 5 in the updated appendices.

Overall, the paper is well-written and addresses an important clinical issue. Minor improvements in discussion clarity, consistency of presentation, and language use will enhance the manuscript’s clarity and inclusivity. Well done

Reviewer 2 Report

Comments and Suggestions for Authors

The study presents a systematic review and meta-analysis of VR in relation to needling procedures in cancer patients.

It is a well-written manuscript that complies with PRISMA guidelines. I only have a few suggestions for improvement.

If there are discrepancies, how are they resolved? Line 148 says one reviewer, and line 157 says two reviewers.

Improve the figure in the table “Reports sought for retrieval” and “Reports excluded with reasons” as the number is not easy to read.

Why is the first article in Table 1 in yellow?

Although it is not the objective of the study, a brief explanation of how VR is applied or the types of VR in the introduction would be interesting for readers.

Similarly, differentiating the application/explaining the differences between children and adults could be interesting.

Were there any adverse effects, such as dizziness, in any of the articles included? It would be good to mention this limitation.

Conflicts of interest sections are duplicated.

Author Response

#Reviewer 2

The study presents a systematic review and meta-analysis of VR in relation to needling procedures in cancer patients.

It is a well-written manuscript that complies with PRISMA guidelines. I only have a few suggestions for improvement. 

  1. If there are discrepancies, how are they resolved? Line 148 says one reviewer, and line 157 says two reviewers.

A1: Thank you for your inquiry. We acknowledge the inconsistency in the manuscript regarding resolution of discrepancies. The processes of cross-screening and data extraction were performed by two independent reviewers (DJ and KYJ). Any discrepancies or disagreements were resolved through consultation with two additional reviewers (WWR and ZYC). In response to your question, we have revised the relevant section of the manuscript to improve clarity and consistency, please see Line 204-206.

  1. Improve the figure in the table “Reports sought for retrieval” and “Reports excluded with reasons” as the number is not easy to read.

A2: Thank you for your suggestion. We have improved the figure to enhance the readability of the numbers under “reports sought for retrieval” and “reports excluded with reasons”. Please see the improved figure in the revised manuscript.

  1. Why is the first article in Table 1 in yellow?

A3: Thank you for your question. We acknowledge the inappropriate highlighting of the first article in Table 1. We have removed the highlight in the revised manuscript.

  1. Although it is not the objective of the study, a brief explanation of how VR is applied or the types of VR in the introduction would be interesting for readers.

A4: Thank you for your suggestion. We have added a brief explanation of the applications and type of VR in the introduction section to enhance reader understanding. Please see the revised manuscript Line 107-111.

  1. Similarly, differentiating the application/explaining the differences between children and adults could be interesting.

A5: Thank you for your suggestion. We have also included a brief explanation of the differences in VR applications between children and adults. Please refer to the revised manuscript Line 111-114.

  1. Were there any adverse effects, such as dizziness, in any of the articles included? It would be good to mention this limitation.

A6: Thank you so much for your question. We have addressed the potential adverse effects of virtual reality, such as cybersickness and digital eye strain, in the limitation section. Please refer to the revised manuscript, see Line 141-143.

  1. Conflicts of interest sections are duplicated.

A7: Thank you for pointing this out, we have removed the duplicate in the revised manuscript. 

Reviewer 3 Report

Comments and Suggestions for Authors

This article systematically evaluates the effects of virtual reality (VR)-based interventions against psychological (anxiety, depression, fear) and physiological (pain, respiratory/pulse rate) symptoms seen during needle-related procedures in cancer patients.

1) P values ​​should be given as small p in the text. For significance, p<0.001 is sufficient. P < 0.00001 is written in many places.
2) It would be appropriate if GRADE results were given visually.
3) If there is a PROSPERO record, it should be added; if not, the reason why it was not added should be discussed.

Author Response

#Reviewer 3

This article systematically evaluates the effects of virtual reality (VR)-based interventions against psychological (anxiety, depression, fear) and physiological (pain, respiratory/pulse rate) symptoms seen during needle-related procedures in cancer patients.

  1. P values ​​should be given as small p in the text. For significance, p<0.001 is sufficient. P < 0.00001 is written in many places.

A1: Thank you for your valuable suggestion. We have replaced all instances of “P” values with lowercase “p” throughout the text. Additionally, we have revised all values reported as P < 0.00001 and P < 0.0001 to p < 0.001, in accordance with your recommendation.

  1. It would be appropriate if GRADE results were given visually.

A2: Thank you for your advice. The GRADE results have been presented visually in Appendix 3 of the revised manuscript.

  1. If there is a PROSPERO record, it should be added; if not, the reason why it was not added should be discussed.

A3: Thank you for your inquiry. The protocol for this systematic review was prospectively registered in PROSPERO with registration ID CRD42025615497. We have added this information to the Methods section of the revised manuscript, please see Line 169-170.
